# Attenuated Total Reflectance-Fourier Transform Infrared (ATR-FTIR) Spectroscopy Discriminates the Elderly with a Low and High Percentage of Pathogenic CD4+ T Cells

**DOI:** 10.3390/cells11030458

**Published:** 2022-01-28

**Authors:** Rian Ka Praja, Molin Wongwattanakul, Patcharaporn Tippayawat, Wisitsak Phoksawat, Amonrat Jumnainsong, Kanda Sornkayasit, Chanvit Leelayuwat

**Affiliations:** 1Biomedical Sciences Program, Graduate School, Khon Kaen University, Khon Kaen 40002, Thailand; riankapraja@kkumail.com; 2The Centre for Research and Development of Medical Diagnostic Laboratories (CMDL), Faculty of Associated Medical Sciences, Khon Kaen University, Khon Kaen 40002, Thailand; moliwo@kku.ac.th (M.W.); patchatip@kku.ac.th (P.T.); amonrat@kku.ac.th (A.J.); kandas@kkumail.com (K.S.); 3Department of Clinical Microbiology, Faculty of Associated Medical Sciences, Khon Kaen University, Khon Kaen 40002, Thailand; 4Department of Microbiology, Faculty of Medicine, Khon Kaen University, Khon Kaen 40002, Thailand; wisiph@kku.ac.th; 5Research and Diagnostic Center for Emerging Infectious Diseases, Khon Kaen University, Khon Kaen 40002, Thailand; 6Department of Clinical Immunology and Transfusion Sciences, Faculty of Associated Medical Sciences, Khon Kaen University, Khon Kaen 40002, Thailand

**Keywords:** aging, attenuated total reflectance-Fourier transform infrared (ATR-FTIR) spectroscopy, interleukin (IL)-17, immunosenescence, sub-population CD4+ T cells

## Abstract

In the aging process, the presence of interleukin (IL)-17-producing CD4+CD28-NKG2D+T cells (called pathogenic CD4+ T cells) is strongly associated with inflammation and the development of various diseases. Thus, their presence needs to be monitored. The emergence of attenuated total reflectance-Fourier transform infrared (ATR-FTIR) spectroscopy empowered with machine learning is a breakthrough in the field of medical diagnostics. This study aimed to discriminate between the elderly with a low percentage (LP; ≤3%) and a high percentage (HP; ≥6%) of pathogenic CD4+CD28-NKG2D+IL17+ T cells by utilizing ATR-FTIR coupled with machine learning algorithms. ATR spectra of serum, exosome, and HDL from both groups were explored in this study. Only exosome spectra in the 1700–1500 cm^−1^ region exhibited possible discrimination for the LP and HP groups based on principal component analysis (PCA). Furthermore, partial least square-discriminant analysis (PLS-DA) could differentiate both groups using the 1700–1500 cm^−1^ region of exosome ATR spectra with 64% accuracy, 69% sensitivity, and 61% specificity. To obtain better classification performance, several spectral models were then established using advanced machine learning algorithms, including J48 decision tree, support vector machine (SVM), random forest (RF), and neural network (NN). Herein, NN was considered to be the best model with an accuracy of 100%, sensitivity of 100%, and specificity of 100% using serum spectra in the region of 1800–900 cm^−1^. Exosome spectra in the 1700–1500 and combined 3000–2800 and 1800–900 cm^−1^ regions using the NN algorithm gave the same accuracy performance of 95% with a variation in sensitivity and specificity. HDL spectra with the NN algorithm also showed excellent test performance in the 1800–900 cm^−1^ region with 97% accuracy, 100% sensitivity, and 95% specificity. This study demonstrates that ATR-FTIR coupled with machine learning algorithms can be used to study immunosenescence. Furthermore, this approach can possibly be applied to monitor the presence of pathogenic CD4+ T cells in the elderly. Due to the limited number of samples used in this study, it is necessary to conduct a large-scale study to obtain more robust classification models and to assess the true clinical diagnostic performance.

## 1. Introduction

The aging process affects the function of various organ systems as well as the immune system [1,2]. Age-related changes in the immune system, known as immunosenescence, are characterized by decreased immune responses leading to susceptibility to infectious diseases, increased expression of pro-inflammatory cytokines contributing to inflammation-related diseases, decreased vaccination response, and increased risk of autoimmune events [3,4,5,6]. Furthermore, multiple age-related alterations can occur in the immune system (both innate and adaptive immune systems) [7,8].

In the adaptive immune system, age-related alterations in CD4+ T cell functions include inappropriate T helper subset differentiation, diminished proliferative capacity, and an increase in regulatory T cells [9,10]. Moreover, a notable alteration of CD4+ T cell phenotype is characterized by loss of CD28, which is one of the hallmarks of immunosenescence [11,12]. Previously, the frequency of CD4+CD28- T cells was significantly correlated with age and, in individuals older than 65 years, the percentage of these cells could reach 50% of the total CD4+ T cells [11,13]. Similarly, in pathological conditions, CD4+CD28- T cells aberrantly expressed surface molecules. One of them was the natural killer group 2 member D (NKG2D) receptor [14,15]. Normally, CD4+ T cells possess a negative expression of NKG2D [16]. A study conducted by Phoksawat et al. (2016) reported that a subpopulation of IL-17-producing CD4+CD28-NKG2D+ T cells (identified as pathogenic CD4+ T cells) existed in the circulation of subjects with type 2 diabetes mellitus (T2DM) [17]. A recent study also reported that these pathogenic CD4+ T cells were expanded in the elderly and could produce high levels of IL-17 and IFN-γ [18,19]. Evidently, this cell subpopulation may associate with the pathogenesis, development, and severity of many diseases, including T2DM [17] and cardiovascular diseases [19]. In aging, these cells may contribute to low-grade inflammation and development of diseases; therefore, their presence in the elderly urgently needs to be monitored.

Serum and its soluble bio components are pivotal elements defined as major sources of biomarkers for monitoring health conditions [20,21,22,23]. Exosomes are nano-sized extracellular vesicles (EV) secreted by different types of cells that circulate in biofluids [24,25,26]. Exosomes contain various biochemical components (proteins, lipids, and nucleic acids) that play an important role as a source of biomarkers [23,26,27]. Moreover, the biochemical components within the exosome are determined by the physiological and pathological states of exosome-secreting cells [28]. In the event of immunosenescence, there is a multitude of cellular immune changes that lead to the alteration of exosome bio components [29,30]. It is also likely that high-density lipoprotein (HDL) is a significant component in serum that can be explored as a source of biomarkers [22,31,32]. Proteomics studies have identified more than 85 HDL proteins that play a role in lipid transport and metabolism, hemostasis, metal binding, vitamin transport, and immune response [33]. Aging conditions are always accompanied by low-grade inflammation [34]. Under inflammatory conditions, it has been reported that biochemical components of HDL underwent biochemical changes, especially ApoA-1, a protein responsible for lipid transport. Furthermore, the liver expresses some acute phase substances such as serum phospholipase A_2_ (sPLA_2_) and serum amyloid A (SAA) that displace ApoA-I, ApoA-II, and other enzymes in HDL, leading to decreased lipid transport, anti-oxidant, and anti-inflammation capacity [35].

Attenuated total reflectance-Fourier transform infrared (ATR-FTIR) spectroscopy has emerged as a powerful tool in the medical diagnostic field [36,37]. ATR-FTIR has been extensively explored to identify molecular vibrations of biomolecules in various biological samples that are useful for monitoring health status [38,39]. Some advantages provided by ATR-FTIR include the ease of handling samples with relatively short measurement duration (only a few minutes), a small amount of required sample volume, a reagent-free approach, and high signal to noise ratio output that facilitates chemometric analysis. In addition, a single scan of the sample can provide information on the infrared spectrum associated with biomarkers of various diseases [40,41].

The combination of vibrational spectroscopy and machine learning algorithms has been utilized as a diagnostic tool for various diseases with excellent discrimination results. Chatchawal et al. (2021) explored the potential of ATR-FTIR combined with several machine learning algorithms to identify cholangiocarcinoma from human sera. In their study, they used various machine learning methods, such as partial least square discriminant analysis (PLS-DA), support vector machine (SVM), random forest (RF), and neural network (NN). Furthermore, the NN algorithm showed the best performance with a test sensitivity of 80–100% and specificity of 83–100% [42]. In addition, the application of ATR-FTIR combined with a neural network could differentiate between sera from healthy subjects and from breast cancer patients with a sensitivity of 92–95% and specificity of 95–100% [43]. Another finding reported the possibility of ATR-FTIR coupled with deep learning to be applied for stratifying healthy, allergic, and allergen-specific immuno-therapy in mice and humans [44]. Unfortunately, to the best of our knowledge, ATR-FTIR empowered with machine learning algorithms has not been widely applied to investigate immunosenescence.

In this study, we hypothesized that changes occurring in immunosenescence associated with the presence of CD4+CD28-NKG2D+ T cells producing IL-17 could be observed in spectra of serum, exosome, and HDL. We therefore demonstrated the possible use of machine learning-empowered ATR-FTIR to discriminate the elderly groups with a low percentage (LP; ≤3%) and a high percentage (HP; ≥6%) of these pathogenic CD4+ T cells. In addition to the discrimination, we also investigated biochemical changes in serum, exosome, and HDL samples as well as oxidative stress levels in the two groups.

## 2. Materials and Methods

### 2.1. Samples

Left-over sera were employed in this study. Sera were collected from elderly subjects aged >60 years, and the percentage of IL-17-producing CD4+CD28-NKG2D+ (pathogenic CD4+) T cells from peripheral blood mononuclear cells (PBMCs) was investigated by flowcytometric analysis. Serum collection and flowcytometric analysis were conducted by Sornkayasit et al. (2021) [18]. In this study, 21 and 22 serum samples from subjects with a low percentage (LP; ≤3%) and a high percentage (HP; ≥6%), respectively, of these pathogenic CD4+ T cells were selected. Subjects with a percentage of these pathogenic CD4+ T cells with a range > 3 and <6% were considered as a gray zone group and not included in this study. All sera were aliquoted and stored at −80 °C until use. This study was approved by the Ethics Committee of Khon Kaen University (HE 631335).

### 2.2. HDL Isolation

HDL was isolated using the HDL purification kit (Cell Biolabs, Inc., San Diego, CA, USA, cat no. STA, 607) with modifications referring to Praja et al. (2021) [45]. All solutions used in this step were the same as in the original kit without any modifications. Modifications were only done in terms of the kit solution volume used. Smaller volumes of HDL purification kit and serum samples were employed. Briefly, 1 µL of dextran solution and 10 µL of precipitation solution A were added to 200 µL of serum and then incubated for 5 min on ice. The mixture was later centrifuged at 6000× *g* for 10 min (at 4 °C), and the collected supernatant was transferred to a new tube. Twelve microliters of dextran solution and 30 µL of precipitation solution A were added and incubated for 2 h at room temperature. The mixture was centrifuged at 18,000× *g* for 30 min (at 4 °C). Sequentially, the supernatant was discarded, and a pellet was resuspended using 100 µL of HDL resuspension buffer followed by centrifugation at 6000× *g* for 10 min (at 4 °C). The supernatant was discarded, and the pellet was then mixed with 120 µL of 1X HDL wash solution and shaken for 30 min at 4 °C. The supernatant was transferred to a new sterile tube, and 18 µL of dextran removal solution was added, followed by incubation for 1 h at 4 °C. The mixture was then centrifuged again at 6000× *g* for 10 min (at 4 °C). Finally, the isolated HDL in the supernatant was transferred into a separate new tube and stored at −80 °C prior to use.

### 2.3. Exosome Isolation

Exosome isolation was carried out using ExoQuick™ Exosome Precipitation Solu-tion (System Biosciences, Palo Alto, CA, USA, cat no. EXOQ5A-1) with a modified method according to Praja et al. (2021) [45]. Before use, to remove cell and cell debris, serum samples were centrifuged at 3000× *g* for 15 min. Next, 50 µL of serum was transferred to a new tube and 12.6 µL of ExoQuick™ was added and the mixture was refrigerated for 30 min. The mixture was then centrifuged at 1500× *g* for 30 min. The supernatant was discarded, and the pellet was centrifuged again at 1500× *g* for 5 min. The remaining supernatant was discarded, then the pellet was resuspended using 1× phosphate buffer saline (PBS). The isolated exosomes were stored at −80 °C until use.

### 2.4. FTIR Spectral Acquisition

Sample spectra were recorded using Agilent 4500 FTIR spectroscopy (Agilent Technologies, Santa Clara, CA, USA). Prior to spectrum collection, ATR-FTIR was cleaned using deionized (DI) water and methanol. The background was measured by scanning the diamond ATR surface. For spectrum collection, 3 µL of the sample was placed on the surface of the ATR and dried using a hair dryer for 5 min. The FTIR spectra were obtained at the wavenumber range of 4000–650 cm^−1^ with 64 scans and a spectral resolution of 4 cm^−1^. From each sample, three spectra were collected.

### 2.5. Spectral Band Area Analysis

This approach was done to quantitatively investigate biochemical components of serum, exosome, and HDL samples from the LP and HP groups. Briefly, raw spectra were preprocessed with baseline correction and a Savitsky–Golay algorithm with 3 polynomial orders and 13 smoothing points. The standard normal variate (SNV) normalization was employed to normalize the spectra [46]. Regions of 3000–2800 cm^−1^ (lipid), 1700–1500 cm^−1^ (protein), and 1270-960 cm^−1^ (nucleic acid) were selected for analysis [39,47]. The spectral band area calculation was done with SpectraGryph version 1.2.14 (Dr. Friedrich Menges, Oberstdorf, Germany).

### 2.6. Oxidative Stress Study by FTIR

To investigate oxidative stress levels in both LP and HP groups, lipid peroxidation and protein carbonyl were analyzed. ATR-FTIR serum spectra were employed for this analysis. The same preprocessed serum spectra in the spectral band area analysis were used in this section. Lipid peroxidation was investigated by calculating ratio *v* C=O (1760-1720 cm^−1^) to *vas* CH_3_ (2982–2942 cm^−1^) [48], and protein carbonyl was studied by calculating the ratio of *v* C=O (1760–1720 cm^−1^) to amide II (1585–1480 cm^−1^) [49]. Calculation of the spectral band area ratio was done using SpectraGryph version 1.2.14 (Dr. Friedrich Menges, Oberstdorf, Germany).

### 2.7. Principal Component Analysis (PCA)

The Unscrambler^®^X software version 10.4 for Windows 64 bit (Camo Software AS, Oslo, Norway) was employed for the analysis. Briefly, second derivative spectra were calculated by the Savitzky–Golay algorithm with 13 smoothing points and 3 polynomial orders. To normalize the spectra, data transformation was done using the standard normal variate (SNV). Finally, PCA was generated to observe the trend of discrimination between the LP and HP groups.

### 2.8. Spectral Data Analysis Using Multiple Algorithms

All preprocessed spectra from the LP and HP groups were combined. Furthermore, all spectra were randomized and allocated to either a training set or a testing set with a ratio of 70:30 using Weka software version 3.8.4. (The University of Waikato, Hamilton, New Zealand). The first classification model built was partial least square-discriminant analysis (PLS-DA) employing The Unscrambler^®^X software version 10.4 for Windows 64 bit (Camo Software AS, Oslo, Norway). Various other classification models, including the J48 decision tree, random forest (RF), support vector machine (SVM), and neural network (NN) were created using Weka software version 3.8.4. with the same dataset used in PLS-DA. Specifically, for NN algorithms, hidden layers in the range 1–20 were applied to obtain the best classification model. After creating the classification models, the testing performance of each classification model was measured in terms of % accuracy, sensitivity, specificity, positive predictive value (PPV), and negative predictive value (NPV) (Figure 1).

### 2.9. Classification Model Evaluation

Classification model performance was evaluated in terms of % accuracy, sensitivity, specificity, positive predictive value (PPV), and negative predictive value (NPV) [42,50]. A confusion matrix or 2 × 2 table (Table 1) was employed for calculations with the following formulas:(1)$$\%\;\mathrm{Accuracy}=\left(\frac{A+D}{A+B+C+D}\right)\times 100\;\mathrm{Sensitivity}=\left(\frac{A}{A+C}\right)\times 100$$
(2)$$\%\;\mathrm{Specificity}=\left(\frac{D}{B+D}\right)\times 100\;\%\;\mathrm{PPV}=\left(\frac{A}{A+B}\right)\times 100$$
(3)$$\%\;\mathrm{NPV}=\left(\frac{D}{C+D}\right)\times 100$$

### 2.10. Statistical Analysis

Statistical analyses were performed using SPSS Statistics for Windows, version 17.0 (SPSS Inc., Chicago, IL, USA). Data collected in this study were tested for their distribution by the Shapiro Wilk test. Data related to the integral area of spectra were analyzed using independent sample *t*-test and Mann–Whitney test for normal and non-normal distribution, respectively. Mean with standard error of the mean (SEM) was used for data with normal distribution and median with 95% CI was used for data with non-normal distribution.

## 3. Results

### 3.1. Biomolecular Content Study by Spectral Band Area Analysis

In this study, 21 and 22 serum samples from elderly subjects with a low percentage (LP; ≤3%) and a high percentage (HP; ≥6%), respectively, of IL-17-producing CD4+CD28-NKG2D+ T cells were selected. Exosome and HDL were then isolated from sera. ATR spectra from serum, exosome, and HDL samples were collected and then compared in terms of biochemical contents between the LP and HP groups. We calculated the spectral band area of the region 3000–2800 cm^−1^ (lipid), 1700–1500 cm^−1^ (protein), and 1270–960 cm^−1^ (nucleic acid) (Figure 2A).

The spectral band area of serum, exosome, and HDL spectra was calculated with SpectraGryph v.1.2.14 (Dr. Friedrich Menges, Oberstdorf, Germany). All spectra were preprocessed with baseline correction, the Savitsky–Golay 13 smoothing points with 3 polynomial orders, and the standard normal variate (SNV) normalization. We found that lipid, protein, and nucleic acid contents in the sera of the LP and HP subjects were not significantly different (*p* > 0.05). According to the spectral band area analysis of exosome spectra, it was identified that lipid and nucleic acid contents in the HP group were significantly higher than in the LP group (*p* < 0.001 and *p* < 0.01, respectively). In contrast, no significant difference was shown by the protein spectral band area of exosome spectra (*p* > 0.05). Additionally, the spectral band area on the protein of HDL was significantly different between the HP and LP groups (*p* < 0.05). However, no significant differences were identified in terms of lipid and nucleic acid contents of HDL (*p* > 0.05) (Figure 2B–D).

### 3.2. Oxidative Stress Study

The levels of oxidative stress were investigated by calculating spectral band area of ratio *v* C=O (1760–1720 cm^−1^) to *vas* CH3 (2982–2942 cm^−1^) corresponding to lipid peroxidation (Figure 3A) and the ratio of *v* C=O (1760–1720 cm^−1^) to amide II (1585–1480 cm^−1^) corresponding to protein carbonyl (Figure 3B). Data preprocessing were baseline correction and Savitsky–Golay with 13 smoothing points and 3 polynomial orders. In addition, the standard normal variate (SNV) was used to normalize the spectra. Analysis was done with SpectraGryph v.1.2.14 (Dr. Friedrich Menges, Oberstdorf, Germany).

The study of oxidative stress levels by using serum spectra showed that lipid peroxidation and protein carbonyl levels were not statistically different between the LP and HP groups (*p* > 0.05). However, the trend of the data showed that the LP group had lower oxidative stress based on the ratio *v* C=O to *vas* CH_3_ and *v* C=O to amide II.

### 3.3. Differences between Spectra from the LP and HP Groups

In Figure 4, we present the absorbance bands mainly in lipids (3000–2800 cm^−1^) (pale-pink box) and mixed region of protein/phospholipids/DNA/RNA/carbohydrate (1800–900 cm^−1^) (pale-grayish green box). All averaged second derivative spectra exhibited similar spectral patterns for all regions by visual examination of the raw spectra. However, some peaks were identified to have a different intensity in the exosome and HDL spectra. The average of second derivative ATR-FTIR spectra with SNV normalization of the serum, exosome, and HDL spectra for the spectral regions from 3000–2800 and 1800–900 cm^−1^ are shown.

By visual examination in the regions of 3000–2800 and 1800–900 cm^−1^, it was identified that the averaged spectral pattern of the LP and HP groups in serum spectra was similar (Figure 4A), whereas the averaged spectra of exosomes in the 1800–900 cm^−1^ region had some different band intensities based on visual examination. However, in the 1800–900 cm^−1^ region of exosome spectra, only the band at 1107 cm^−1^ had significantly higher intensity in the HP group as compared to the LP group (*p* < 0.05) (Figure 4B). A band at 1107 cm^−1^ was assigned to *v* (CO) and *v* (CC). Through a visual observation in the HDL spectra, peaks 1058 and 984 cm^−1^ in the fingerprint 1800–900 cm^−1^ region seemed higher in the LP group compared to the HP group. Furthermore, the independent sample *t*-test revealed a significant difference only at the band 984 cm^−1^ assigned to uracil ring motions of RNA [51] (Figure 4C).

### 3.4. Discrimination by Unsupervised Analysis: Principal Component Analysis (PCA)

A total of 129 serum, exosome, and HDL spectra from the elderly with different percentages of IL-17-producing CD4+CD28-NKG2D+ T cells (LP; ≤3% versus HP; ≥6%) were analyzed by principal component analysis (PCA) for identifying whether any clustering of spectra could be observed and discriminated in each group. Spectra regions used for PCA were lipid region (3000–2800 cm^−1^), mixed region (1800–900 cm^−1^), protein region (1700–1500 cm^−1^), and fingerprint region (1500–900 cm^−1^). Among all regions of the serum, exosome, and HDL spectra analyzed using PCA, most of the regions employed could not show discrimination between the two groups as exhibited by representative score plots (Appendix A). However, the wave-number around 1700–1500 cm^−1^ of exosome spectra showed a possible separation between the LP and HP groups (Figure 5). These results showed that the protein region could be used to discriminate between the LP and HP groups. Additionally, a score plot representing most of the LP and HP groups were separated along the PC-1 (59%) (Figure 5A).

Analysis of the PCA loading plots was used to determine the regions of the FTIR spectrum which contributed the most to the clustering. Bands at 1670, 1629, and 1558 cm^−1^ had a contribution to clustering the LP group in the PC-1 positive score plot. Bands at 1651 and 1541 cm^−1^ were important for the HP group (Figure 5B).

### 3.5. Establishment of Partial Least Square Discriminant Analysis (PLS-DA) Model for Discrimination

According to PCA analysis, the trend of classification from exosome spectra was only found in the protein region (1700–1500 cm^−1^). The PLS-DA analysis was then explored for generating a possible classification model. Herein, exosome spectra were divided into two datasets, a training set and a testing set, with a ratio of 70:30. A training set consisting of 90 spectra was modeled by PLS-DA to produce a predictive model. The remaining 39 spectra employed as a testing set were predicted utilizing a generated PLS model with the 1700–1500 cm^−1^ region. The performance of the generated PLS model was evaluated in terms of % accuracy, sensitivity, specificity, positive predictive value (PPV), and negative predictive value (NPV). The PLS model generated using the 1700–1500 cm^−1^ region gave a discrimination along with Factor-1 (x-axis) (Figure 6A). The regression coefficients exhibited three important bands for the LP group, including 1670, 1626, and 1555 cm^−1^ (blue box). Furthermore, wavenumber values at 1651 and 1541 cm^−1^ were significant for the HP group (pink box) (Figure 6B).

Based on the PLS-DA model, exosome spectra from the HP group were set to a positive value (+1), whereas the LP group was assigned to a negative value (−1) (Figure 6C). Regarding the predictive model, 14 false predictions consisting of nine false-negative and five false-positive predictions were observed. The performance of the PLS-DA model run in the 1700–1500 cm^−1^ region exhibited 64% accuracy, 69% sensitivity, 61% specificity, 55% PPV, and 74% NPV as calculated based on confusion matrix (2 × 2 table) (Appendix A).

As the PCA had a limitation in generating the classification model and the performance of the PLS-DA model was not favorable enough to discriminate two groups of the elderly, we further attempted to employ advanced machine learning algorithms to obtain better classification results. For this further work, we utilized J48 decision tree, random forest (RF), support vector machine (SVM), and neural network (NN) to generate better classification models.

### 3.6. Classification Model Using Advanced Machine Learning Algorithms

In an attempt to create better classification models, we employed several advanced machine learning algorithms, including J48 decision tree, random forest (RF), support vector machine (SVM), and neural network (NN). The classification models were then generated using five spectral ranges: 3000–2800, 1800–900, 1700–1500, 1500–900, and combined 3000–2800 and 1800–900 cm^−1^. A ratio of 70:30 was selected to split the dataset (70% for the training set and 30% for the testing set). The performance of classification models was evaluated in terms of % accuracy, sensitivity, specificity, positive predictive value (PPV), and negative predictive value (NPV).

Overall, the analysis to discriminate between the elderly groups with LP and HP of pathogenic CD4+CD28-NKG2D+IL17+ T cells using multiple advanced machine learning algorithms gave better results than the PCA and PLS-DA models. The summary of discrimination results in serum, exosome, and HDL spectra using multiple algorithms are shown in Table 2, Table 3 and Table 4.

Based on the PCA and PLS-DA, separation between the LP and HP groups in serum spectra from the five spectral regions were not found. Therefore, some advanced machine learning approaches were employed to classify the LP and HP groups. According to Table 2, the 3000–2800 cm^−1^ spectral range of serum spectra was an unfavorable region for generating classification models because it had very low accuracy values even though we utilized several advanced machine learning approaches. Classification models of serum spectra developed using the J48 decision tree algorithm had an accuracy in the range of 54–74%. The best performance of classification models by the J48 decision tree algorithm was in the combined region of 3000–2800 and 1800–900 cm^−1^ with accuracy, sensitivity, and specificity of 74, 81, and 70%, respectively. Models by random forest (RF) provided better performance than the J48 decision tree with the best accuracy, sensitivity, and specificity of 92, 100, and 86%, respectively, in the 1800–900 cm^−1^ spectral region. We then further created classification models based on the support vector machine (SVM) algorithm. The best performance of the classification model by SVM was in the spectral range 1800–900 cm^−1^ with 77% accuracy, 79% sensitivity, and 75% specificity. Interestingly, in the 1800–900, 1700–1500, 1500–900, and the combined 3000–2800 and 1800–900 cm^−1^ regions, classification models based on neural network (NN) possessed very high accuracy in the range of 90–100%. Predominantly, the performance of the classification model by NN in the 1800–900 cm^−1^ region exhibited a value of 100% for all parameters.

Of the five exosome spectral regions that we used to differentiate between the LP and HP groups based on PCA, only the 1700–1500 cm^−1^ protein region showed possible discrimination between the two groups. Unfortunately, the accuracy, sensitivity, and specificity given by the classification model by PLS-DA were 64, 69, and 61%, respectively. The results of PLS-DA in the 1700–1500 cm^−1^ region might not be satisfied although we could observe the discrimination. We, therefore, explored the use of advanced machine learning algorithms (J48 decision tree, RF, SVM, and NN) to classify the LP and HP groups. Referring to Table 3, all spectral regions of the exosome spectra had good potential to be used to differentiate between the LP and HP groups. Various classification models created using various algorithms in the 3000–2800 cm^−1^ region showed an accuracy of 72, 74, 77, and 85% for SVM, RF, NN, and J48 decision tree, respectively. In the other regions, 1800–900, 1700–500, 1500–900, and 3000–2800 and 1800–900 cm^−1^, classification models by the NN algorithm with different hidden layers provide varying accuracy of approximately 90–95%. Interestingly, the 1700–1500 cm^−1^ region and the combined region of 3000–2800 and 1800–900 cm^−1^ generated with different hidden layers of the NN algorithm showed the same accuracy performance of 95%. Specifically, the best-hidden layer of the NN algorithm for the classification model in the spectra region 1700–1500 cm^−1^ was 11; in the combination region, the NN algorithm with nine hidden layers was the best parameter. In addition, there were PPV and NPV differences between the classification models built based on the NN algorithm in the 1700–1500 cm^−1^ spectral region and the combination of 3000–2800 and 1800–900 cm^−1^. Other algorithms used, such as RF and SVM, provided the potential for differentiating LP and HP groups with varying accuracy ranging from 74–90% and 72–82% for RF and SVM, respectively.

According to Table 4, classification models of HDL spectra with various advanced machine learning algorithms at wavenumber 3000–2800 cm^−1^ showed an unsatisfactory potential for discrimination between the LP and HP groups, with the best accuracy of 72% by the NN with eight hidden layers. Classification models generated using SVM and RF had good performance to differentiate between the LP and HP groups with >70% ac-curacy in some spectral regions. However, the worst performance of the classification model by RF was in the spectral region 3000–2800 cm^−1^ with an accuracy of 44%. The lowest performance of SVM was in the HDL 1700–1500 cm^−1^ spectral range with an accuracy of 51%. Interestingly, the classification models based on HDL spectra revealed that the NN algorithm with a variety of hidden layers was the best algorithm for generating better classification models in the various regions used. The accuracy performance shown by the NN algorithm was 72–97%. A classification model based on the 1800–900 cm^−1^ region generated using the NN with 14 hidden layers gave the best model performance possessing an accuracy, sensitivity, specificity, PPV, and NPV of 97, 100, 95, 95, and 100%, respectively. Notably, the classification models in the region of 1500–900 cm^−1^ created using the NN and J48 decision tree algorithms showed similar accuracy performance of 92%, but different in terms of sensitivity, specificity, PPV, and NPV.

## 4. Discussion

This study explored the ATR-FTIR combined with machine learning algorithms to classify elderly subjects with different percentages of IL-17-producing CD4+CD28-NKG2D+ T cells. For the LP group, the subjects were individuals with a percentage of these pathogenic T cells of ≤3% (*n* = 21), whereas for the HP group, individuals with a percentage ≥ 6% were selected (*n* = 22). Subjects with a percentage of pathogenic CD4+ T cells ranging from >3 to <6% were not recruited (considered as a gray zone). Serum, exosome, and HDL from both LP and HP groups were employed to collect ATR spectra. This study is the first evidence in the field of immunosenescence discriminating the elderly with LP and HP of IL-17-producing CD4+CD28-NKG2D+ T cells by utilizing machine learning-empowered ATR-FTIR.

First, we analyzed the biochemical components in each sample of the LP and HP groups. The spectral band area analysis of serum samples showed no significant difference in lipids, proteins, and nucleic acids in the two groups (*p* > 0.05) (Figure 2B). Based on the spectral band area analysis of exosome spectra, the nucleic acid of the HP group was significantly higher than that of the LP group (*p* < 0.01). The components of nucleic acids carried by exosomes include microRNA, long noncoding RNA, circular RNA, and DNA [52]. Under systemic inflammation, pro-inflammatory exosomes are secreted and circulate in the bloodstream, leading to inflammation of distant tissue. Moreover, those pro-inflammatory exosomes were documented to carry various pro-inflammatory microRNAs, including miR-15a, miR-27b, and miR-125a [53]. Therefore, the increase of nucleic acid contents in exosomes might be correlated with the increase of pro-inflammatory miRNAs. According to the statistical analysis, lipid levels in the exosome of LP group subjects were significantly lower when compared to the exosome of HP group subjects (*p* < 0.001) and there was no significant difference between the exosome proteins of the LP and HP groups (*p* > 0.05) (Figure 2C). The different lipid levels in exosomes of the two groups might be related to the immune status during inflammation. Generally, exosomes are capable of transporting lipids, such as cholesterol, fatty acids, as well as eicosanoids from parent cells to recipient cells, leading to inflammation or changes in immunity and metabolism [54]. A study by Kakazu et al. (2016) demonstrated that pro-inflammatory exosomes were enriched in ceramide (a sphingolipid family member) in the study of non-alcoholic steatohepatitis (NASH) [55]. Another study also reported that ceramide and dihydroceramide in cystic fibrosis extracellular vesicles (CF-EVs) were higher than in control EVs [56]. Furthermore, excessive levels of ceramide could lead to inflammation [56,57]. As previously explained, the exosome component depends on exosome-secreting cells [28]. In inflammatory diseases, such as systemic lupus erythematosus (SLE), peripheral blood T cells isolated from subjects with SLE had higher cholesterol and glycosphingolipid GM1 (a lipid raft biomarker) levels in the plasma membrane than did healthy individuals. The study proved that activated T cells in SLE subjects produce more cholesterol and GM1 [58]. Furthermore, lipid components in the outer and inner layers of the exosome membrane are expected to have similarities with the plasma membrane [59]. Cholesterol is the main lipid component of exosomes, with levels around 42–63% depending on the parent cells [60]. However, because exosomes carry different types of lipids [60], in this study we could not identify exactly which lipid components were elevated in the exosome of the HP group.

For the HDL spectral band area analysis, no significant difference was found in the lipid and nucleic acid contents (*p* > 0.05). Specifically, the protein content in HDL was significantly higher in the HP group than that the LP group (*p* < 0.05) (Figure 2D). CD4+CD28-NKG2D+ T cells producing IL-17 have been documented in inflammation-associated diseases, including T2DM [17] and cardiovascular diseases [19]. During inflammation, HDL protein biocomponents undergo alterations as the liver produces more acute phase substances such as serum phospholipase A2 (sPLA2) and serum amyloid A (SAA) that displace ApoA-I, ApoA-II, and other enzymes in HDL, causing reduction of antioxidant, anti-inflammatory, and lipid transport capacity [35]. In addition, ceruloplasmin was also identified to be increased in dysfunctional HDL. Furthermore, in those with chronic disease characterized by inflammation and oxidative stress, HDL may promote inflammation response [61]. Further research involving omics approaches (transcriptomics, proteomics, and lipidomics) may be useful to deeply investigate biochemical differences between LP and HP groups. Therefore, the exact biochemical differences could be obtained. According to the lipid peroxidation and serum carbonyl analyses, the levels of oxidative stress between the LP and HP groups were not statistically different (*p* > 0.05), although the LP group tended to have lower levels of oxidative stress (Figure 3A,B).

Before performing the discrimination using unsupervised analysis and advanced machine learning algorithms, we attempted to visually analyze the difference of spectra from the LP and HP groups, employing the second derivative spectra (Figure 4). Unfortunately, the patterns of the second derivative spectra of serum, exosome, and HDL from the two groups were similar. The differences were only found in the band intensity of exosome (1107 cm^−1^ assigned to *v* (CO) and *v* (CC)) and HDL (984 cm^−1^ assigned to uracil ring motions of RNA) [51]. Those differences in band intensities were not enough to classify the LP and HP groups. The principal component analysis (PCA) and partial least square-discriminant analysis (PLS-DA) uncovered several important bands from spectra of exosomes to discriminate between the LP and HP groups (Table 5). The prominent spectral bands and proposed biomolecular assignments for the HP group were at bands 1651 cm^−1^ (α-helix amide I) and 1541 cm^−1^ (amide II). Bands at 1670, 1626, and 1555/8 cm^−1^ were bands associated with the LP group. A band at 1670 was a vibration of amide I (anti-parallel β-sheet), *v* (C=C) trans, lipids, and fatty acids. Spectral peaks at 1629 and 1626 cm^−1^ were assigned to β-sheet amide I region structure. The last bands at 1555/8 consisted of ring base vibration [51,62,63,64,65]. In general, the spectral peaks associated with the LP and HP discrimination from exosome spectra were associated with differences in protein, lipid, and nucleic acid biomolecules. In agreement with our findings, several studies employing exosome bio component analyses also found that proteins, lipids, and nucleic acids underwent biochemical composition alterations under various inflammation mediated diseases [66,67,68].

Evidently, IL-17-producing CD4+CD28-NKG2D+ T cells were considered as one of the contributors to inflammation in aging [18]. Thus, their existence may be additionally monitored in the elderly. Generally, the presence of these pathogenic CD4+ T cells can be investigated using flowcytometric analysis [17,18]. Although the flowcytometric technique is commonly performed, it requires expensive antibodies, special tools, and more complicated procedures. Furthermore, studies using FTIR combined with machine learning approaches to study immunosenescence were poorly studied. In this study, ATR-FTIR spectroscopy coupled with machine learning algorithms was developed to study these particular CD4+ T cells by exploring serum, exosome, and HDL from the LP and HP groups. Previously, a study of FTIR in the immunological field, specifically allergy, attempting to classify healthy, allergic, and allergen-specific immuno-therapy was conducted by Korb et al. (2020). Their classification models of the FTIR spectra of human sera generated using deep learning showed an overall accuracy of 93.9%. Furthermore, the model they established successfully discriminated against allergy, allergen-specific immunotherapy, and healthy individuals with true positive rates of 93.3, 91.7, and 96.7%, respectively [44]. Another FTIR study classifying inflammatory fibrous hyperplasia (IFH) lesions and normal oral mucosa (NM) using the PCA-LDA approach had a sensitivity and specificity of 87.5 and 100%, respectively [69]. Our classification models using various types of advanced machine learning algorithms, including PLS-DA, J48 decision tree, random forest (RF), support vector machine (SVM), and neural network (NN) provided a variety of diagnostic performances. The classification models generated by the NN algorithm resulted in the best performance with an accuracy of 100% in serum (1800–900 cm^−1^), 95% in exosomes (1700–1500 and 3000–2800 and 1800–900 cm^−1^), and 97% in HDL (1800–900 cm^−1^). The different performances of a classification model in terms of % accuracy, sensitivity, and specificity are not only influenced by the algorithm used, but also the input attributes, data complexity, and sample size. Until now, there has been no agreement on the best method of analysis [42]. A limitation of this study was the number of spectra used for building the classification models. Ideally, to build a robust classification model using SVM and NN, more than 1,000 samples are required [70,71,72,73]. However, in this study, only 129 spectra were employed. Thus, to obtain more robust classification models and to assess the real classification model performance, a large-scale study is still needed to increase the sample size.

This study showed the advantages of ATR-FTIR spectroscopy combined with multivariate analysis and several advanced machine learning algorithms to classify the LP and HP groups. Based on PCA and PLS-DA, exosomes are the most likely source of biomarkers. Particularly, the application of advanced machine learning algorithms exploring all types of samples (serum, exosomes, and HDL) could be used to classify these two groups. Conclusively, ATR-FTIR may be one of the effective alternative tools that can be used to study changes in the immune system in aging. Additionally, ATR-FTIR may be suitable for studying multiple biochemical alterations in biological samples where a single FTIR spectrum can provide various biochemical information related to health conditions.

## 5. Conclusions

In this work, we presented that machine learning-assisted ATR-FTIR spectroscopy could be effectively applied to investigate an immunological alteration in immunosenescence. ATR-FTIR combined with advanced machine learning algorithms allows differentiating the elderly with a low percentage (LP) and a high percentage (HP) of IL-17-producing CD4+CD28-NKG2D+ T cells from serum, exosome, and HDL samples with favorable performances. However, to improve the robustness of SVM- and NN-based classification models and to assess the real diagnostic performance, a large-scale study with a larger sample size is still needed.

## Figures and Tables

**Figure 1 cells-11-00458-f001:**
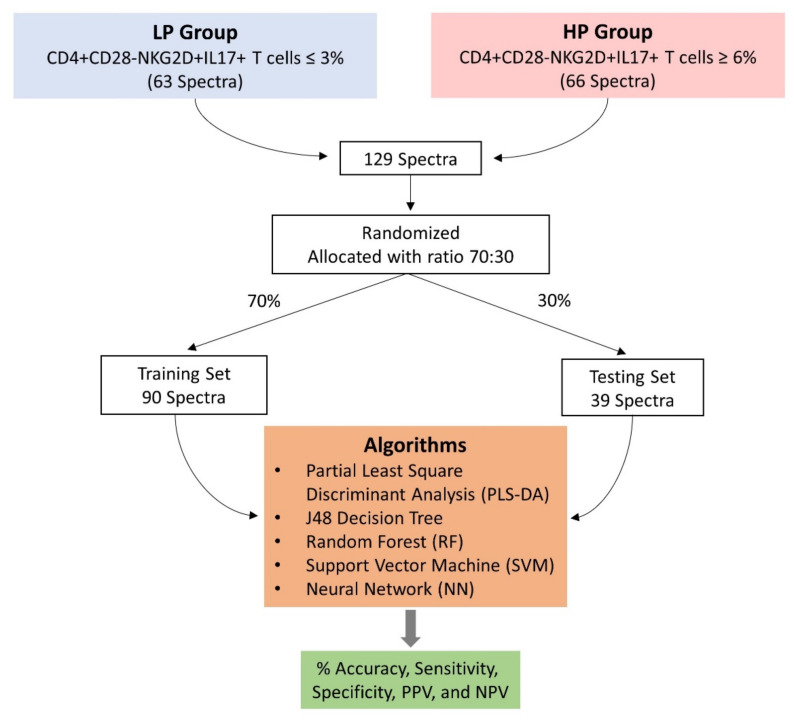
Workflow of the development of classification models and their performance tests. All spectra collected from two groups of samples were allocated with a ratio of 70:30 into two datasets for training and testing. The training set was used for creating classification models and the performance of the classification models was tested with a testing set.

**Figure 2 cells-11-00458-f002:**
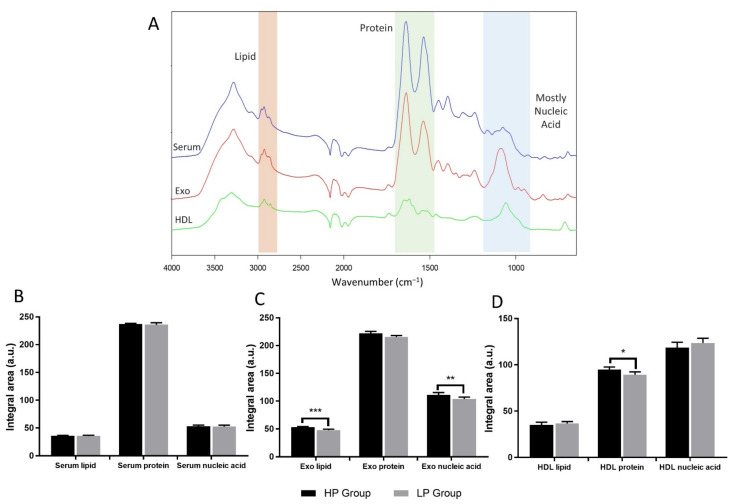
Results of spectral band area analysis of serum, exosome, and HDL spectra. Representative spectra from serum (blue), exosome (red), and HDL (green). Regions selected for the spectral band area analysis were 3000–2800 cm^−1^ lipid (orange region), 1700–1500 cm^−1^ protein (green region), and 1270–960 cm^−1^ nucleic acid (blue region) (**A**). The comparison of spectral band area between the LP and HP groups based on serum (**B**), exosome (**C**), and HDL spectra (**D**); * = *p* < 0.05; ** = *p* < 0.01; *** = *p* < 0.001.

**Figure 3 cells-11-00458-f003:**
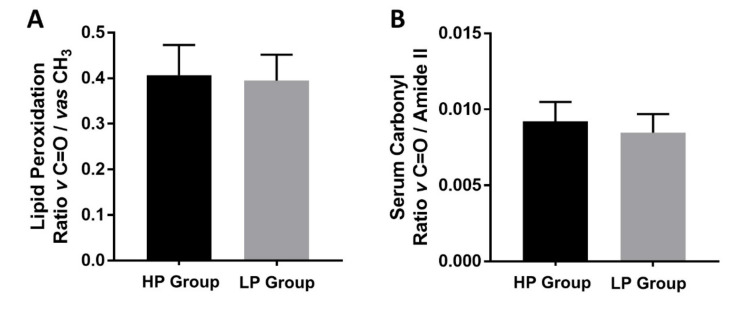
Oxidative stress levels based on the ratio of the spectral band area. A ratio of *v* C=O to vas CH_3_ (lipid peroxidation) (**A**) and the ratio of *v* C=O to amide II (protein carbonyl) (**B**) All data are shown as median with 95% CI.

**Figure 4 cells-11-00458-f004:**
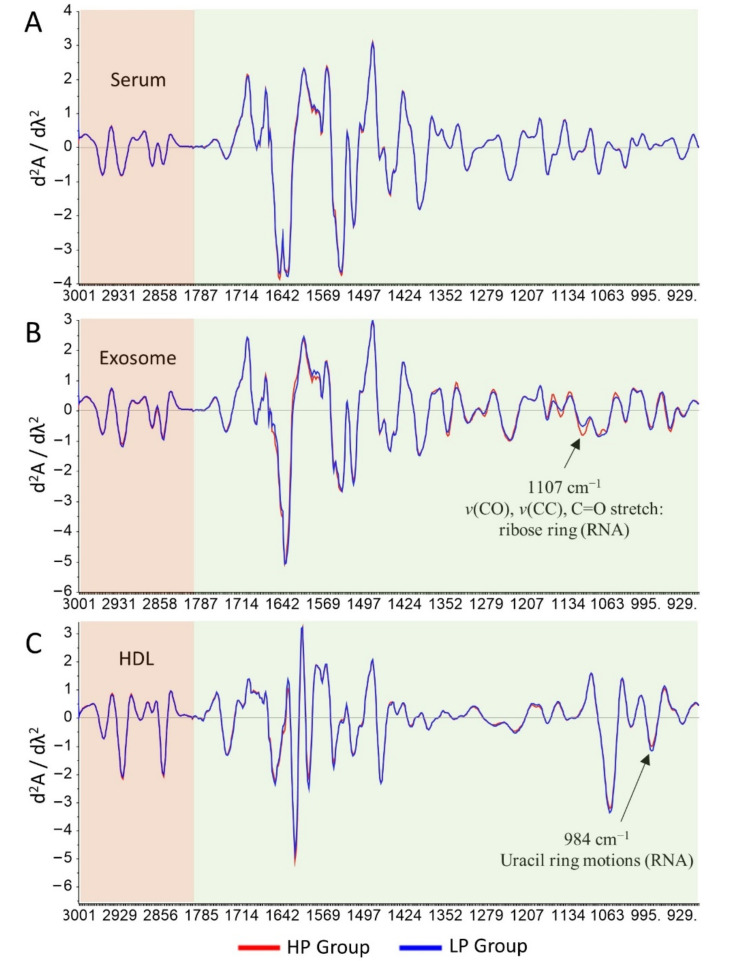
Averaged second derivative ATR-FTIR spectra with SNV normalization in the regions of 3000–2800 cm^−1^ (pale-pink box) and 1800–900 cm^−1^ (pale-grayish green box). Comparison of averaged second derivative spectra between LP and HP groups in serum (**A**), exosome (**B**), and HDL spectra (**C**). Comparison of band intensity was done by independent sample *t*-test. A significant difference in band intensity is depicted by the arrow (→). Blue and red represent the elderly groups with a low percentage (LP; ≤3%) and a high percentage (HP; ≥6%) of pathogenic CD4+ T cells, respectively.

**Figure 5 cells-11-00458-f005:**
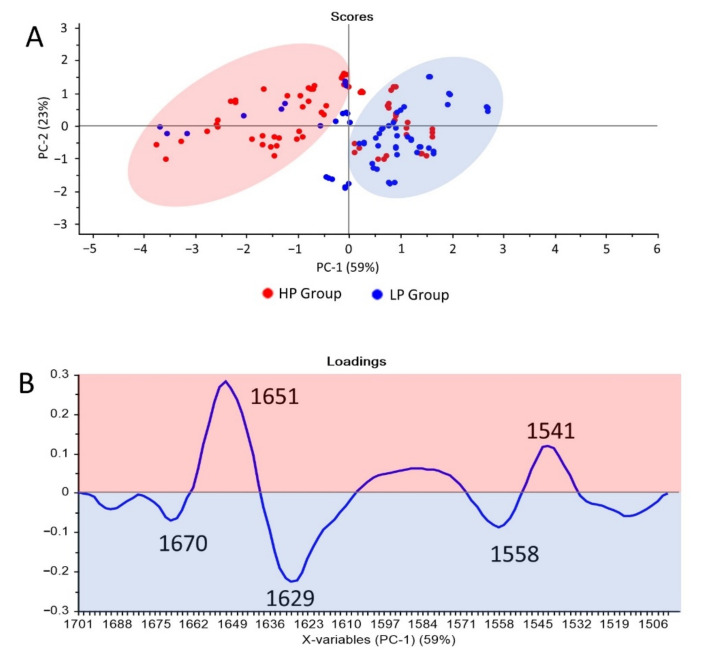
PCA analysis of the 1700–1500 cm^−1^ FTIR exosome spectral range. PCA score plots (**A**) and PCA loading plots (**B**). PCA score plots showed distinct clustering between the LP (blue box) and HP groups (pink box). PCA loading plots identify specific important peaks for the LP and HP groups.

**Figure 6 cells-11-00458-f006:**
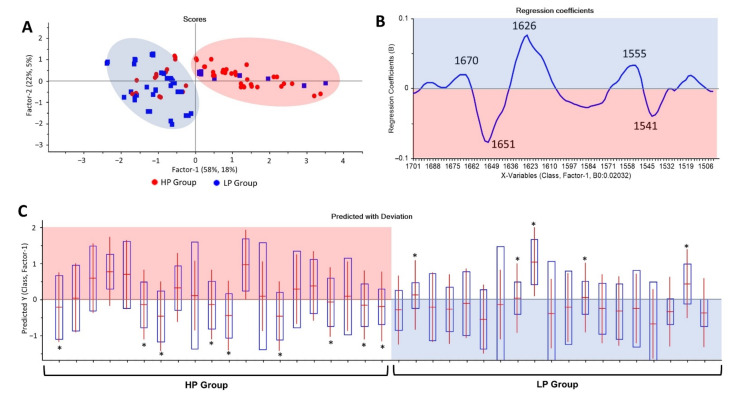
PLS-DA analysis results. A score plot of PLS-DA of the 1700–1500 cm^−1^ FTIR exosome spectral range (**A**), regression coefficient (**B**), and predictive results of PLS-DA generated using the 1700–1500 cm^−1^ region (**C**). False predictions are depicted with stars (*). Nine false-negative and five false-positive predictions were identified with the PLS-DA predictive model.

**Table 1 cells-11-00458-t001:** The confusion matrix (2 × 2 table).

Predictive Model	Flowcytometric Analysis	
	HP	LP
**HP**	A	B
**LP**	C	D

**Table 2 cells-11-00458-t002:** Comparison of multiple advanced machine learning algorithms for classification models in serum samples.

Sample	Region (cm^−1^)	Algorithm	Performance				
			Acc (%)	Sens (%)	Spec (%)	PPV (%)	NPV (%)
Serum	3000–2800	J48 Decision Tree	54	54	53	65	42
		RF	51	53	50	50	53
		SVM	44	46	38	60	26
		NN (4)	51	53	50	50	53
	1800–900	J48 Decision Tree	72	80	67	60	84
		RF	92	100	86	85	100
		SVM	77	79	75	75	79
		NN (20)	100	100	100	100	100
	1700–1500	J48 Decision Tree	69	75	65	60	79
		RF	90	90	89	90	89
		SVM	62	58	75	90	32
		NN (14)	90	86	94	95	84
	1500–900	J48 Decision Tree	56	60	54	45	68
		RF	90	90	89	90	89
		SVM	72	74	70	70	74
		NN (12)	97	100	95	95	100
	3000–2800 and 1800–900	J48 Decision Tree	74	81	70	65	84
		RF	87	89	85	85	89
		SVM	56	58	55	55	58
		NN (11)	98	95	100	100	97

Abbreviations: Acc—accuracy; Sens—sensitivity; Spec—specificity; PPV—positive predictive value; NPV—negative predictive value; RF—random forest; SVM—support vector machine; NN—neural network. Values in the parentheses after NN indicate the number of hidden layers used in the NN parameter. Values highlighted in grey are the best model in each spectral region.

**Table 3 cells-11-00458-t003:** Comparison of multiple advanced machine learning algorithms for classification models in exosome samples.

Sample	Region (cm^−1^)	Algorithm	Performance				
			Acc (%)	Sens (%)	Spec (%)	PPV (%)	NPV (%)
Exosome	3000–2800	J48 Decision Tree	85	79	93	95	74
		RF	74	75	74	75	74
		SVM	72	71	72	75	68
		NN (14)	77	79	75	75	79
	1800–900	J48 Decision Tree	82	81	83	85	79
		RF	90	90	89	90	89
		SVM	74	81	70	65	84
		NN (10)	90	94	86	85	95
	1700–1500	J48 Decision Tree	67	67	67	70	63
		RF	79	83	76	75	84
		SVM	72	76	68	65	79
		NN (11)	95	95	95	95	95
	1500–900	J48 Decision Tree	85	94	78	75	95
		RF	87	86	89	90	84
		SVM	72	76	68	65	79
		NN (16)	92	90	94	95	89
	3000–2800 & 1800–900	J48 Decision Tree	79	83	76	75	84
		RF	90	90	89	90	89
		SVM	82	84	80	80	84
		NN (9)	95	91	100	100	89

Abbreviations: Acc—accuracy; Sens—sensitivity; Spec—specificity; PPV—positive predictive value; NPV—negative predictive value; RF—random forest; SVM—support vector machine; NN—neural network. Values in the parentheses after NN indicate the number of hidden layers used in the NN parameter. Values highlighted in grey were the best model in each spectral region.

**Table 4 cells-11-00458-t004:** Comparison of multiple advanced machine learning algorithms for classification models in HDL samples.

Sample	Region (cm^−1^)	Algorithm	Performance				
			Acc (%)	Sens (%)	Spec (%)	PPV (%)	NPV (%)
HDL	3000–2800	J48 Decision Tree	69	79	64	55	84
		RF	44	45	42	45	42
		SVM	56	56	57	70	42
		NN (8)	72	70	75	80	63
	1800–900	J48 Decision Tree	72	74	70	70	74
		RF	85	85	84	85	84
		SVM	74	73	76	80	68
		NN (14)	97	100	95	95	100
	1700–1500	J48 Decision Tree	79	83	76	75	84
		RF	74	75	74	75	74
		SVM	51	52	50	60	42
		NN (8)	79	77	82	85	74
	1500–900	J48 Decision Tree	92	95	90	90	95
		RF	90	83	100	100	79
		SVM	77	74	81	85	68
		NN (9)	92	100	86	85	100
	3000–2800 & 1800–900	J48 Decision Tree	90	94	86	85	95
		RF	82	84	80	80	84
		SVM	69	67	73	80	58
		NN (15)	90	100	83	80	100

Abbreviations: Acc—accuracy; Sens—sensitivity; Spec—specificity; PPV—positive predictive value; NPV—negative predictive value; RF—random forest; SVM—support vector machine; NN—neural network. Values in the parentheses after NN indicate the number of hidden layers used in the NN parameter. Values highlighted in grey were the best model in each spectral region.

**Table 5 cells-11-00458-t005:** Prominent ATR-FTIR exosome spectral bands for discrimination of the LP and HP groups using PCA and PLS-DA [51,62,63,64,65].

PCA Band (cm^−1^)	PLS-DA Band (cm^−1^)	Group	Assignment
1651	1651	HP	Amide I (α-helix)
1541	1541	HP	Amide II
1670	1670	LP	Amide I (anti-parallel β-sheet) *v* (C=C) trans, lipids, and fatty acids
1629	1626	LP	β-sheet amide I region structure
1558	1555	LP	Ring base

## Data Availability

Not applicable.

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
