# Peer review of "Attenuated Total Reflectance-Fourier Transform Infrared (ATR-FTIR) Spectroscopy Discriminates the Elderly with a Low and High Percentage of Pathogenic CD4+ T Cells"

_cells, 2022, doi:10.3390/cells11030458_

Round 1

Reviewer 1 Report

In the following manuscript: Attenuated Total Reflectance‐Fourier Transform Infrared (ATR‐FTIR) Spectroscopy Discriminates the Elderly with a Low and High Percentage of Pathogenic CD4+ T Cells, Praja et al. utilized ATR-FTIR coupled with a number of further informatics analyses of sera, exosomes and HDL particles to discriminate between samples originated from subjects with high or low content of CD4+CD28-NKG2D+ T cells.

Overall rationale of the study is sound, the methods and the literature presented are appropriate. Results are presented in comprehensive and convincing way leading to the well-grounded conclusion. However, I have 2 comments that could improve the clarity and correctness of the study.

Firstly, in HDL isolation (lines 135-150), please provide contents of solutions used (solution A and resuspension buffer). Since the protocol is slightly changed in comparison to the one proposed by manufacturer, and is described briefly, for the sake of clarity it would be necessary to provide detailed content or at least statement that those solutions are the same as from the original kit (for example in the brackets where the solutions are mentioned or otherwise).

Secondly, in Amide I analyses, 1651 cm-1 band is attributed to Amide I maximum, I suppose (line 521 and Table 5) while 1629 and 1670 are attributed to beta-sheets. Although It can be found in the some literature to be attributed this way, in secondary derivate spectrum analysis this band (1651) correspond to alpha-helix (beside beta-sheets, another protein secondary structure). Citations can be found everywhere in the literature (alpha-helix spans in the Amide I region approximately within 1645-1655). Depending on protein secondary structures content in the samples, amid I maximum can shift towards slightly lower and higher frequencies, but in secondary derivate spectrum is not present. In secondary derivate spectrum, bands contributing to Amide I as a whole are resolved, thus 1651 should be attributed to Amide I alpha-helix band (the authors already did this for beta-sheets bands). Attribution of 1541 to Amide II as a whole region is correct since (in contrary to Amide I), Amide II is not very sensitive to protein secondary structures and thus is not resolved to its constituents by secondary derivate calculation.

Author Response

Reviewer 1

In the following manuscript: Attenuated Total Reflectance‐Fourier Transform Infrared (ATR‐FTIR) Spectroscopy Discriminates the Elderly with a Low and High Percentage of Pathogenic CD4+ T Cells, Praja et al. utilized ATR-FTIR coupled with a number of further informatics analyses of sera, exosomes and HDL particles to discriminate between samples originated from subjects with high or low content of CD4+CD28-NKG2D+ T cells.

Overall rationale of the study is sound, the methods and the literature presented are appropriate. Results are presented in comprehensive and convincing way leading to the well-grounded conclusion. However, I have 2 comments that could improve the clarity and correctness of the study.

Comment 1

Firstly, in HDL isolation (lines 135-150), please provide contents of solutions used (solution A and resuspension buffer). Since the protocol is slightly changed in comparison to the one proposed by manufacturer, and is described briefly, for the sake of clarity it would be necessary to provide detailed content or at least statement that those solutions are the same as from the original kit (for example in the brackets where the solutions are mentioned or otherwise).

Response:

Thanks for the very constructive comments and suggestions. When we submitted our previous work entitled “Alternative Method for HDL and Exosome Isolation with Small Serum Volumes and Their Characterizations” to Separations (MDPI), the reviewer also commented and questioned about the formula of the HDL purification kit (Cell Biolabs, Inc., San Diego, CA, USA, cat no. STA, 607). We then discussed with the representative of the company producing the kit. Unfortunately, it is the proprietary of the company. Therefore, the company's representative could not inform us the components of the kit.

During the process of HDL isolation, we did not modify the components of each solution of  the HDL purification kit. We only modified the kit volume used. We used smaller serum sample volume followed by proportional reduction of the kit volume used. Morover, the results of this modified method has been published in Separations (https://www.mdpi.com/2297-8739/8/11/204).

We have added an extra explanation on page 3, line 137-140 to read "All solutions used in this step were the same as the original kit without any modifications. Modifications were only done in terms of kit solution volume used. Smaller volumes of HDL purification kit and serum samples were employed.”

Comment 2

Secondly, in Amide I analyses, 1651 cm-1 band is attributed to Amide I maximum, I suppose (line 521 and Table 5) while 1629 and 1670 are attributed to beta-sheets. Although It can be found in the some literature to be attributed this way, in secondary derivate spectrum analysis this band (1651) correspond to alpha-helix (beside beta-sheets, another protein secondary structure). Citations can be found everywhere in the literature (alpha-helix spans in the Amide I region approximately within 1645-1655). Depending on protein secondary structures content in the samples, amid I maximum can shift towards slightly lower and higher frequencies, but in secondary derivate spectrum is not present. In secondary derivate spectrum, bands contributing to Amide I as a whole are resolved, thus 1651 should be attributed to Amide I alpha-helix band (the authors already did this for beta-sheets bands). Attribution of 1541 to Amide II as a whole region is correct since (in contrary to Amide I), Amide II is not very sensitive to protein secondary structures and thus is not resolved to its constituents by secondary derivate calculation.

Response:

We are very grateful with your suggestion. We made 1651 cm-1 more detail to be Amide I (α-helix).  We have modified the sentence on page 19, line 597-599 to read “The prominent spectral bands and proposed biomolecular assignments for the HP group were at bands 1651 cm-1 (α-helix amide I) and 1541 cm-1 (amide II).”

We have also added additional supporting references, Corbalán-García, 2003 and Ye at al., 2017, that mentioned 1651 cm-1 is attributed to alpha-helix amide I [1,2]. The additional supporting references were on page 20, line 603 (now ref no 64-65) and on the Table 5, page 20. Reference appearances in the manuscript were auto-arranged according to the order in which they appeared.

References

  1. Corbalán-García, S.; García-García, J.; Sánchez-Carrillo, M.S.; Gómez-Fernández, J.C. Structural Characterization of the C2 Domains of Classical Isozymes of Protein Kinase C and Novel Protein Kinase Cε by Using Infrared Spectroscopy. Spectroscopy 2003, 17, 399–416, doi:10.1155/2003/361563.
  2. Ye, M.P.; Zhou, R.; Shi, Y.R.; Chen, H.C.; Du, Y. Effects of Heating on the Secondary Structure of Proteins in Milk Powders Using Mid-Infrared Spectroscopy. J. Dairy Sci. 2017, 100, 89–95, doi:10.3168/jds.2016-11443.

Reviewer 2 Report

Presented manuscript is interesting and a lot of high quality test were done to verify the hypothesis. However I have, several questions and comments

  1. What was the volume of the sample, which were put to the ATR crystal?
  2. How long the samples were dried? It is very important, because drying for too long causes the crystallization process to take place.
  3. Were the spectra normalized before calculating the half area under the peaks? If so, what kind of normalization was used? These information should be included in the manuscript. Moreover, without normalization, the obtained results may not be reliable due to e.g. different volumes of the measured samples.

Author Response

Reviewer 2

Presented manuscript is interesting and a lot of high quality test were done to verify the hypothesis. However I have, several questions and comments

Comment 1

What was the volume of the sample, which were put to the ATR crystal?

Response:

Thank you for your constructive comments. We used 3 µl of serum, exosome, and HDL samples. We have added this explanation in the section “2.4. FTIR Spectral Acquisition” on page 4 line 171-172 to read “For spectrum collection, 3 µL of the sample was placed on the surface of the ATR and dried using a hair drier for 5 minutes.”

Comment 2

How long the samples were dried? It is very important, because drying for too long causes the crystallization process to take place.

Response:

Each sample was dried for 5 minutes. We also have added this extra information in the section “2.4. FTIR Spectral Acquisition” on page 4 line 171-172 to read “For spectrum collection, 3 µL of the sample was placed on the surface of the ATR and dried using a hair drier for 5 minutes.”

Comment 3

Were the spectra normalized before calculating the half area under the peaks? If so, what kind of normalization was used? These information should be included in the manuscript. Moreover, without normalization, the obtained results may not be reliable due to e.g. different volumes of the measured samples.

Response:

As we used the same volume of samples (3 µl) and the same duration of drying process (5 minutes), therefore, previously we only used baseline correction and the Savitsky-Golay algorithm with 13 smoothing points and 3 polynomial orders to prepare the spectra for the spectral area analysis. However, after we got your suggestion regarding the reliability of the results, we considered to re-analyze our spectra by adding the Standard Normal Variate (SNV) method to normalize our data before conducting the spectral area analysis.

We would like to emphasize that we conducted separated analyses between the spectral area analysis and the discrimination of the HP and LP groups using multiple algorithms. Thus, our re-analysis in the part of spectral area analysis does not affect to our main/major results related to the discrimination of the elderly with a high and low percentage of pathogenic CD4+ T cells.

There are some changes that has been made after re-analysis including:

  1. In the Materials and Methods section “2.5 Spectral Band Area Analysis” on page 4 line 180-181 to read “The Standard Normal Variate (SNV) normalization was employed to normalize the spectra [1].
  2. In the Materials and Methods section “2.6 Oxidative Stress Study by FTIR” on page 4 line 189-190 to read “The same preprocessed serum spectra in the spectral band area analysis were used in this section.”
  3. As we re-analyzed the spectral band area analysis, thus, Figure 2 has been changed as shown on page 9.
  4. A paragraph explaining Figure 2 has also been modified on page 7, line 265-277 to read “The spectral band area of serum, exosome, and HDL spectra was calculated with SpectraGryph v.1.2.14 (Dr. Friedrich Menges, Oberstdorf, Germany). All spectra were preprocessed with baseline correction, the Savitsky-Golay 13 smoothing points with 3 polynomial orders, and the Standard Normal Variate (SNV) normalization. We found that lipid, protein, and nucleic acid contents in the serum of the LP and HP subjects were not significantly different (p > 0.05). According to the spectral band area analysis of exosome spectra, it was identified that lipid and nucleic acid contents in the HP group were significantly higher than in the LP group (p < 0.001 and p < 0.01, respectively). On the other hand, no significant difference was shown by the protein spectral band area of exosome spectra (p > 0.05). Additionally, the spectral band area on protein of HDL was significantly different between the HP and LP groups (p < 0.05). However, no significant differences were identified in terms of lipid and nucleic acid contents of HDL (p > 0.05) (Figure 2B, 2C, and 2D).”
  5. As we re-analyzed the oxidative stress, thus, Figure 3 has also been changed as shown on page 10.
  6. A sentence has been added in the paragraph explaining Figure 3 on page 9, line 296-297 to read “In addition, the Standard Normal Variate (SNV) was used to normalize the spectra.”
  7. In the section of Discussion, as there were some changes in our results, we modified our discussion on page 17-18, line 539-589 to read “Firstly, we analyzed the biochemical components in each sample of the LP and HP groups. The spectral band area analysis of serum samples showed no significant difference in lipids, proteins, and nucleic acids in the two groups (p > 0.05) (Figure 2B). Based on the spectral band area analysis of exosome spectra, the nucleic acid of the HP group was significantly higher than that of the LP group (p < 0.01). The components of nucleic acids carried by exosomes include microRNA, long noncoding RNA, circular RNA, and DNA [2]. Under systemic inflammation, pro-inflammatory exosomes are secreted and circulate in the bloodstream leading to inflammation to distant tissue. Moreover, those pro-inflammatory exosomes were documented to carry various pro-inflammatory microRNAs, including miR-15a, miR-27b, and miR-125a [3]. Therefore, the increase of nucleic acid contents in exosome might be correlated with the increase of pro-inflammatory miRNAs. According to the statistical analysis, lipid levels in the exosome of LP group subjects were significantly lower when compared to the exosome of HP group subjects (p < 0.001) and there was no significant difference between the exosome proteins of the LP and HP groups (p > 0.05) (Figure 2C). The different lipid levels in exosomes of the two groups might be related to the immune status during inflammaging. Generally, exosomes are capable of transporting lipids, such as cholesterol, fatty acids as well as eicosanoids from parent cells to recipient cells, leading to inflammation or changes in immunity and metabolism [4]. A study by Kakazu et al. (2016) demonstrated that pro-inflammatory exosomes were enriched in ceramide (a sphingolipid family member) in the study of non-alcoholic steatohepatitis (NASH) [5]. Another study also reported that ceramide and dihydroceramide in cystic fibrosis extracellular vesicles (CF-EVs) were higher than control EVs [6]. Furthermore, excessive levels of ceramide could lead to inflammation [6,7]. As previously explained, the exosome component depends on exosome-secreting cells [8]. In inflammatory diseases, such as systemic lupus erythematosus (SLE), peripheral blood T cells isolated from subjects with SLE had higher cholesterol and glycosphingolipid GM1 (a lipid raft biomarker) levels in the plasma membrane than healthy individuals. The study proves that activated T cells in SLE subjects produce more cholesterol and GM1 [9]. Furthermore, lipid components in the outer and inner layers of the exosome membrane are expected to have similarities with the plasma membrane [10]. Cholesterol is the main lipid component of exosomes with levels around 42-63% depending on the parent cells [11]. However, because exosomes carry different types of lipids [11], therefore, in this study we could not identify exactly which lipid components were elevated in the exosome of the HP group. For the HDL spectral band area analysis, no significant difference was found in the lipid and nucleic acid contents (p > 0.05). Specifically, the protein content in HDL was significantly higher in the HP group than that the LP group (p < 0.05) (Figure 2D). CD4+CD28-NKG2D+ T cells producing IL-17 have been documented present in inflammation-associated diseases, including T2DM [12] and cardiovascular diseases [13]. During inflammation, HDL protein biocomponents undergoes alterations as liver produces more acute phase substances such as serum phospholipase A2 (sPLA2) and serum amyloid A (SAA) that displace ApoA-I, ApoA-II, and other enzymes in HDL causing reduction of anti-oxidant, anti-inflammatory, and lipid transport capacity [14]. In addition, ceruloplasmin was also identified to be increased in dysfunctional HDL. Furthermore, in those with chronic disease characterized by inflammation and oxidative stress, HDL may promote inflammation response [15]. A further research involving omics approaches (transcriptomics, proteomics, and lipidomics) may be useful to deeply investigate biochemical differences between LP and HP groups. Therefore, the exact biochemical differences could be obtained. According to the lipid peroxidation and serum carbonyl analyses, the levels of oxidative stress between the LP and HP groups were not statistically different (p > 0.05) although the LP group tended to have lower levels of oxidative stress (Figure 3A and 3B).

Reference appearances in the manuscript were auto-arranged according to the order in which they appeared.

References

  1. Lebon, M.; Reiche, I.; Gallet, X.; Bellot-Gurlet, L.; Zazzo, A. Rapid Quantification of Bone Collagen Content by ATR-FTIR Spectroscopy. Radiocarbon 2016, 58, 131–145, doi:10.1017/RDC.2015.11.
  2. Gusachenko, O.N.; Zenkova, M.A.; Vlassov, V.V. Nucleic Acids in Exosomes: Disease Markers and Intercellular Communication Molecules. Biochem. Mosc. 2013, 78, 1–7, doi:10.1134/S000629791301001X.
  3. Awadasseid, A.; Wu, Y.; Zhang, W. Extracellular Vesicles (Exosomes) as Immunosuppressive Mediating Variables in Tumor and Chronic Inflammatory Microenvironments. Cells 2021, 10, 2533, doi:10.3390/cells10102533.
  4. Record, M.; Carayon, K.; Poirot, M.; Silvente-Poirot, S. Exosomes as New Vesicular Lipid Transporters Involved in Cell-Cell Communication and Various Pathophysiologies. Biochim. Biophys. Acta 2014, 1841, 108–120, doi:10.1016/j.bbalip.2013.10.004.
  5. Kakazu, E.; Mauer, A.S.; Yin, M.; Malhi, H. Hepatocytes Release Ceramide-Enriched pro-Inflammatory Extracellular Vesicles in an IRE1α-Dependent Manner. J. Lipid Res. 2016, 57, 233–245, doi:10.1194/jlr.M063412.
  6. Zulueta, A.; Peli, V.; Dei Cas, M.; Colombo, M.; Paroni, R.; Falleni, M.; Baisi, A.; Bollati, V.; Chiaramonte, R.; Del Favero, E.; et al. Inflammatory Role of Extracellular Sphingolipids in Cystic Fibrosis. Int. J. Biochem. Cell Biol. 2019, 116, 105622, doi:10.1016/j.biocel.2019.105622.
  7. Chaurasia, B.; Talbot, C.L.; Summers, S.A. Adipocyte Ceramides—The Nexus of Inflammation and Metabolic Disease. Front. Immunol. 2020, 11, 2282, doi:10.3389/fimmu.2020.576347.
  8. Im, K.; Baek, J.; Kwon, W.S.; Rha, S.Y.; Hwang, K.W.; Kim, U.; Min, H. The Comparison of Exosome and Exosomal Cytokines between Young and Old Individuals with or without Gastric Cancer. Int. J. Gerontol. 2018, 12, 233–238, doi:https://doi.org/10.1016/j.ijge.2018.03.013.
  9. Jury, E.C.; Kabouridis, P.S. T-Lymphocyte Signalling in Systemic Lupus Erythematosus: A Lipid Raft Perspective. Lupus 2004, 13, 413–422, doi:10.1191/0961203304lu1045rr.
  10. Skotland, T.; Sagini, K.; Sandvig, K.; Llorente, A. An Emerging Focus on Lipids in Extracellular Vesicles. Adv. Drug Deliv. Rev. 2020, 159, 308–321, doi:10.1016/j.addr.2020.03.002.
  11. Skotland, T.; Sandvig, K.; Llorente, A. Lipids in Exosomes: Current Knowledge and the Way Forward. Prog. Lipid Res. 2017, 66, 30–41, doi:10.1016/j.plipres.2017.03.001.
  12. Phoksawat, W.; Jumnainsong, A.; Leelayuwat, N.; Leelayuwat, C. Aberrant NKG2D Expression with IL-17 Production of CD4+ T Subsets in Patients with Type 2 Diabetes. Immunobiology 2017, 222, 944–951, doi:10.1016/j.imbio.2016.05.001.
  13. Phoksawat, W.; Jumnainsong, A.; Sornkayasit, K.; Srisak, K.; Komanasin, N.; Leelayuwat, C. IL-17 and IFN-γ Productions by CD4+ T Cells and T Cell Subsets Expressing NKG2D Associated with the Number of Risk Factors for Cardiovascular Diseases. Mol. Immunol. 2020, 122, 193–199, doi:10.1016/j.molimm.2020.04.003.
  14. Norata, G.D.; Pirillo, A.; Ammirati, E.; Catapano, A.L. Emerging Role of High Density Lipoproteins as a Player in the Immune System. Atherosclerosis 2012, 220, 11–21, doi:10.1016/j.atherosclerosis.2011.06.045.
  15. G, H.B.; Rao, V.S.; Kakkar, V.V. Friend Turns Foe: Transformation of Anti-Inflammatory HDL to Proinflammatory HDL during Acute-Phase Response. Cholesterol 2011, 2011, 274629, doi:10.1155/2011/274629.

Round 2

Reviewer 1 Report

The manuscript is improved and can be accepted in the present form.

Author Response

Thanks

Reviewer 2 Report

I accept this version of manuscript.

Author Response

Thanks
